# VideoMark: A Distortion-Free Robust Watermarking Framework for Video Diffusion Models

## Abstract

This work introduces **VideoMark**, a distortion-free robust watermarking framework for video diffusion models. As diffusion models excel in generating realistic videos, reliable content attribution is increasingly critical. However, existing video watermarking methods often introduce distortion by altering the initial distribution of diffusion variables and are vulnerable to temporal attacks, such as frame deletion, due to variable video lengths. VideoMark addresses these challenges by employing a **pure pseudorandom initialization** to embed watermarks, avoiding distortion while ensuring uniform noise distribution in the latent space to preserve generation quality. To enhance robustness, we adopt a frame-wise watermarking strategy with pseudorandom error correction (PRC) codes, using a fixed watermark sequence with randomly selected starting indices for each video. For watermark extraction, we propose a Temporal Matching Module (TMM) that leverages homology distance to align decoded messages with the original watermark sequence, ensuring resilience against temporal attacks. Experimental results show that VideoMark achieves higher decoding accuracy than existing methods while maintaining video quality comparable to watermark-free generation. The watermark remains imperceptible to attackers without the secret key, offering superior invisibility compared to other frameworks. VideoMark provides a practical, training-free solution for content attribution in diffusion-based video generation.

## 1 Introduction

In recent years, diffusion models have revolutionized the landscape of AI-generated content, emerging as the state-of-the-art technology for image and video generation Ho et al. (2020; 2022); Sohl-Dickstein et al. (2015); Liu et al. (2025a). These models can create highly realistic content that is increasingly indistinguishable from human-created media Rombach et al. (2022b); Huang et al. (2025). The rapid advancement in generation quality has created an urgent need to track and attribute AI-generated content, particularly given growing concerns about copyright infringement and potential misuse Almutairi & Elgibreen (2022); Zhang et al. (2025); Zheng et al. (2026); Pan et al. (2025); Liu et al. (2025b). To address these challenges, watermarking techniques have emerged as a crucial solution for ensuring content traceability and authentication in the era of AI-generated media.

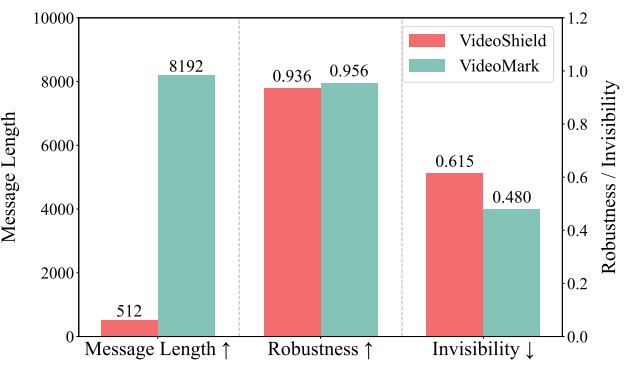

Figure 1: The performance results of VideoMark versus VideoShield across three metrics.

Traditional watermarking methods for both images and videos typically operate as **post-processing techniques**, where watermarks are embedded after content generation Luo et al. (2023); Zhang et al. (2019); Ye et al. (2025). These methods not only require additional computational overhead but also suffer from

limited generalization capabilities. Recent research has shifted towards embedding watermarks during the generation process itself. Leveraging the **reversibility of DDIM** Song et al. (2020), several methods have achieved success in the image domain by manipulating the initial Gaussian noise—e.g., Tree-Ring Wen et al. (2023)—or embedding messages into the noise distribution, as in Gaussian Shading Yang et al. (2024) and PRC-Watermark Gunn et al. (2024).

However, directly adapting image watermarking techniques to the video domain presents unique challenges. First, video DDIM inversion yields **lower accuracy** than image-based methods. And as a result, methods like VideoShield Hu et al. (2025) repeat watermark patterns in initial noise to enhance detection, but these **compromise video quality and watermark invisibility**. Second, watermark robustness suffers against temporal attacks like frame deletion or reordering, because treating videos as a single entity fails to localize watermarks temporally. Furthermore, modifications along the temporal dimension may cause dimensional collapse, whereby the manipulated video no longer conforms to the predefined encoding dimensionality. This misalignment between encoding and decoding spaces ultimately results in watermark decoding failure. Third, real-world videos exhibit variable temporal lengths, whereas many diffusion-based watermarking algorithms rely on fixed noise initialization schemes. This mismatch restricts scalability and limits applicability to practical scenarios.

To address video watermarking challenges, we first define essential characteristics for our proposed watermark. Primarily, the watermark embedded within the initial latent noise must cause negligible perturbation to the original noise space. Secondly, our approach involves inserting unique watermarks into individual frames, ensuring that each frame can be correctly decoded even if the temporal dimension is altered. Building upon this, we establish relative temporal indices across frames to enable accurate reconstruction of the original message sequence during decoding, thus improving robustness.

With these needs in mind, we introduce **VideoMark**, a distortion-free robust watermarking framework designed for video diffusion models. To achieve an imperceptible watermark that preserves the original noise characteristics, VideoMark utilizes pseudorandom error correction (PRC) codes Christ & Gunn (2024). These codes map the watermark bits directly onto the initialized Gaussian noise for every frame. This specific design ensures the watermark integrates seamlessly, thus fulfilling our first design goal. To enable frame-specific watermarking, VideoMark processes each frame's watermark independently while preserving sequential consistency across frames. Specifically, we generate an extended watermark message sequence. For each video, a random starting position within this master sequence initializes the first frame's watermark, and subsequent frames derive their watermarks sequentially. This aligns with our second design objective, facilitating both individualized frame watermarking and temporal coherence.

To accurately extract the watermark, we propose a temporal matching module (TMM), which aligns the decoded message with the embedded watermark sequence, thereby improving decoding accuracy and robustness. Even under temporal attacks such as frame deletion, TMM ensures that the original watermark sequence can still be correctly reconstructed, preserving its robustness.

In our experiments, we evaluate the effectiveness of our watermarking framework across different video diffusion models, demonstrating high decoding accuracy, high-quality generated videos, and strong invisibility. Our watermark achieves higher decoding accuracy compared to VideoShield, which is currently the state-of-the-art watermarking approach for video diffusion models. Additionally, our watermark achieves the best video quality on both the objective video evaluation benchmark VBenchHuang et al. (2024) and subjective assessments, maintaining parity with watermark-free videos. Importantly, our watermark remains undetectable to attackers without the key, ensuring stronger imperceptibility than other watermarking frameworks.

In summary, the contributions of this work are summarized as follows:

- We propose VideoMark, which leverages pseudo-random Gaussian space initialization to achieve undetectable watermarking in video diffusion models.

- We introduce a frame-wise watermarking strategy with extended message sequences, addressing the challenges of variable-length videos and preventing dimensional collapse under temporal attacks.

- Our experiments demonstrate that VideoMark not only improves decoding accuracy over existing methods but also preserves video quality comparable to watermark-free generation across various models and attack conditions.

## 2 Related Work

### 2.1 Video Diffusion Models

Diffusion models Sohl-Dickstein et al. (2015) progressively add noise to map data distributions to a Gaussian prior and recover original data via iterative denoising. Video diffusion models Ho et al. (2022) use a 3D U-Net with interleaved spatial and temporal attention to generate high-quality, temporally consistent frames. Building on latent diffusion models Rombach et al. (2022b), SVD Blattmann et al. (2023) learns a multi-dimensional latent space for high-resolution frame synthesis. During generation, DDIM sampling Song et al. (2021) efficiently reduces sampling steps while maintaining video quality compared to DDPM sampling Ho et al. (2020).

Video diffusion models primarily follow two paradigms: Text-to-VideoWang et al. (2023); Hu et al. (2025); Huang et al. (2024), where videos are generated based on text prompts, and Image-to-VideoZhang et al. (2023a); Hu et al. (2025); Blattmann et al. (2023), where a video is generated starting from a single image. These paradigms enable the generation of realistic videos, but they also raise concerns regarding the potential generation of misleading content or copyright infringement.

### 2.2 Video Watermark

Video watermarking technology embeds imperceptible signals into visual content and leverages dedicated detection algorithms to verify the presence of embedded information Liu et al. (2024). Existing approaches are generally categorized into two paradigms: post-processing schemes and in-processing schemes.

Post-processing schemes introduce minimal visual perturbations, typically at the pixel level. Recent works Fernandez et al. (2024); Luo et al. (2023); Zhang et al. (2019; 2023b) focus on training watermark embedding networks by optimizing discrepancies between watermarked and original videos, as well as encoding-decoding differences. However, these methods may struggle to balance trade-offs between video quality and watermark robustness.

In contrast to post-processing methods, in-processing schemes integrate the watermarking into the video generation process of current generative video models to better utilize their capabilities. For instance, VideoShieldHu et al. (2025) extends the Gaussian Shading techniqueYang et al. (2024) from the image domain to the video domain, achieving improved robustness. However, repeating watermark bits during initialization induces fixed latent patterns, consequently degrading the quality of generated videos. Moreover, VideoShield exhibits limited robustness against temporal manipulations. Its watermarking mechanism relies on a fixed-length latent initialization, implicitly assuming a stable temporal dimension during decoding. Under frame-level operations such as deletion or insertion, this assumption no longer holds, leading to temporal misalignment between the encoded and observed representations. Such misalignment results in dimensional inconsistency and subsequent collapse of the watermark signal, ultimately preventing reliable message recovery from the tampered video. Currently, only one in-processing video watermarking method exists, and prior approaches struggle to balance watermark robustness with video quality. We propose an undetectable video watermarking scheme to address this trade-off.

## 3 Preliminaries

### 3.1 Diffusion Models

Diffusion models generate content through an iterative denoising process. Given a noise schedule $\beta_t t = 1^T$, the forward process gradually adds noise to data $\mathbf{x_0}$:

$$q(\mathbf{x}_t|\mathbf{x}_{t-1}) = \mathcal{N}(\mathbf{x}_t; \sqrt{1 - \beta_t}\mathbf{x}_{t-1}, \beta_t\mathbf{I}) \tag{1}$$

The reverse process is learned to gradually denoise from $\mathbf{x}T \sim \mathcal{N}(0, \mathbf{I})$ to generate content. While DDPM Ho et al. (2020) introduces stochasticity in each denoising step, DDIM Song et al. (2020) provides an approximately invertible deterministic sampling process:

$$\mathbf{x}_{t-1} = \sqrt{\alpha_{t-1}} \left( \frac{\mathbf{x}_t - \sqrt{1 - \alpha_t}\, \boldsymbol{\epsilon}_\theta(\mathbf{x}_t, t)}{\sqrt{\alpha_t}} \right)$$
$$+ \sqrt{1 - \alpha_{t-1}}\, \boldsymbol{\epsilon}_\theta(\mathbf{x}_t, t) \tag{2}$$

This deterministic reversibility enables control over the generation process through manipulation of the initial noise.

### 3.2 Pseudorandom Codes (PRC)

A PRC is a coding scheme that maps messages to statistically random-looking codewords. We adopt the construction from Christ & Gunn (2024), which provides security based on the hardness of the Learning Parity with Noise (LPN) problem.

The PRC framework consists of three core algorithms:

- KeyGen($n, m, \mathrm{fpr}, t$) $\rightarrow$ key: Generates a key for encoding $m$-bit messages into $n$-bit codewords with sparsity parameter $t$

- Encode(key, $\mathbf{m}$) $\rightarrow$ $\mathbf{c}$: Maps message $\mathbf{m}$ to codeword $\mathbf{c} \in \{-1, 1\}^n$

- Decode(key, $\mathbf{s}$) $\rightarrow$ $\mathbf{m}$ or $\emptyset$: Recovers message from potentially corrupted signal $\mathbf{s} \in [-1, 1]^n$

Our implementation supports soft decisions on recovered bits, optimized for robust watermarking (see supplementary materials for details).

### 3.3 Generative Video Watermarking and Threat Model

Diffusion-based video watermarking involves three key functions in the watermarking process:

1. **Generation**: $V = \mathcal{G}(m, k)$, where $\mathcal{G}$ generates a watermarked video $V$ by embedding message $m$ using secret key $k$ during the diffusion process.

2. **Decoding**: $\hat{m} = \mathcal{D}_{PRC}(V)$, where $\mathcal{D}_{PRC}$ extracts the watermark message $\hat{m}$ from the given video $V$.

3. **Detection**: $\{p, d\} = \mathrm{Detect}(m, \hat{m})$, where Detect compares the original message $m$ with the decoded message $\hat{m}$. This function outputs a p-value $p$ and a boolean decision $d$ indicating whether the distance between $m$ and $\hat{m}$ is significantly smaller than that between $m$ and a random message.

We consider active adversaries who may deliberately apply various transformations to the watermarked video in an attempt to remove, distort, or invalidate the embedded watermark. These manipulations include:

- **Temporal Attacks**: Frame dropping, insertion, or reordering, which disrupts the temporal continuity and alters the original frame sequence, leading to misalignment in the temporal structure and potentially degrading the integrity and decodability of the embedded watermark.

- **Spatial Attacks**: Frame-wise manipulations such as Gaussian blurring, colour jittering, and resolution compression, which aim to distort the watermark signal by degrading the visual content of individual frames.

Our framework aims to be robust against these attacks while ensuring the watermark remains imperceptible and the video quality is preserved.

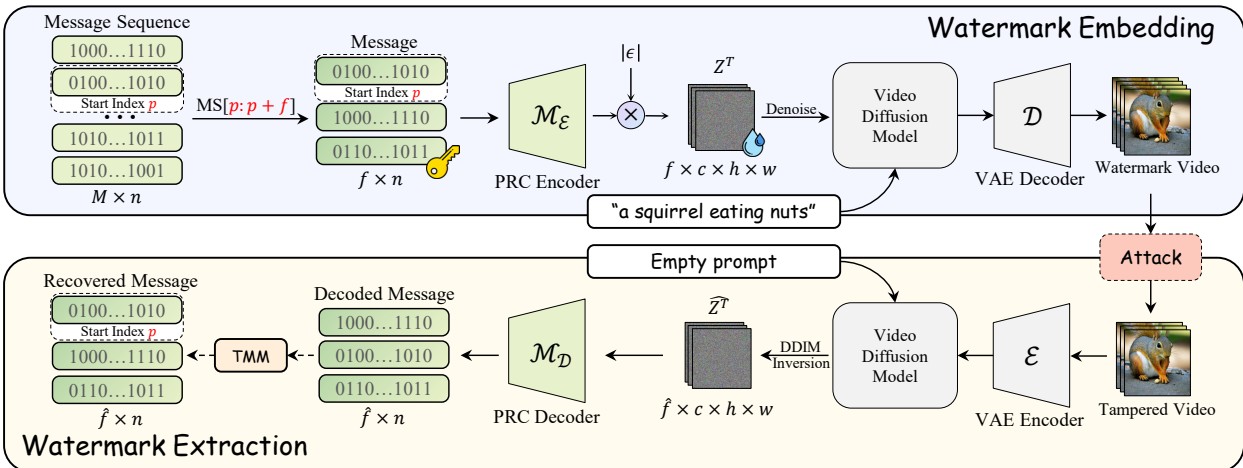

Figure 2: The overall framework of VideoMark. During the watermark embedding phase, $\epsilon$ denotes the standard Gaussian noise sampled randomly. In the I2V task, the first video frame prompts the prediction of initial noise during watermark extraction.

---

**Algorithm 1:** Watermarked Video Generation

**Input** : PRC-key $k$, number of frames $f$, channels $c$, height $h$, width $w$, message $m$, diffusion model $\mathcal{M}$, VAE decoder $\mathcal{D}$

**Output:** Watermarked video $V$

Generate extended message sequence $M$ longer than maximum supported length;
Randomly select starting position $p$ in $M$;
**for** $i = 1$ **to** $f$ **do**
    Extract frame message $m_i$ from $M$ starting at position $p + i$;
    Encode $m_i$ into PRC codeword $c_i \in \{-1, 1\}^{c \times h \times w}$;
    Sample $\epsilon_i \sim \mathcal{N}(0, 1) \in \mathbb{R}^{c \times h \times w}$;
    Compute $\hat{\epsilon}_i \leftarrow c_i \cdot |\epsilon_i|$;

Denoise $\hat{\epsilon}$ using diffusion model $\mathcal{M}$ and decode with VAE decoder $\mathcal{D}$ to obtain video $V$;
**return** $V$;

---

## 4 Proposed Method

In this section, we provide a detailed explanation of the proposed undetectable watermarking method in video diffusion models. Specifically, in Section 4.1, we detail the process of watermark generation. In Section 4.2, we introduce the watermark extraction process.

### 4.1 Watermark Generation

In this section, we detail the watermark generation process. VideoMark achieves superior invisibility and visual quality in diffusion-based watermarking by initializing each frame with pseudo-random Gaussian noise via PRC, followed by DDIM denoising Song et al. (2020) and VAE decoding Kingma & Welling (2013). To ensure diversity and adaptability across variable video lengths, we utilize an extended message list with a randomly initialized start index.

Prior watermarking methods (e.g. VideoShield Hu et al. (2025)) often repeat identical noise patterns, compromising pseudo-randomness, reducing watermark bit capacity, and degrading invisibility and video quality. VideoMark addresses this by generating frame-specific pseudo-random initializations. For a video with $f$ frames, dimensions $c \times h \times w$ (channels, height, width), and a message bit $m'_i$ per frame, the process

is as follows. For each frame $i \in \{1, \ldots, f\}$, we sample Gaussian noise $\boldsymbol{\epsilon}_i \sim \mathcal{N}(0, \mathbf{I}) \in \mathbb{R}^{c \times h \times w}$. Using a PRC key $k$, we encode $m_i'$ to obtain a codeword $\mathbf{c}_i = \text{Encode}(k, m_i') \in \{-1, 1\}^{c \times h \times w}$. The watermarked noise is formally defined as:

$$\hat{\boldsymbol{\epsilon}}_i = \mathbf{c}_i \cdot |\boldsymbol{\epsilon}_i|,$$

where $\mathbf{c}_i$ modulates the sign of $\boldsymbol{\epsilon}_i$, preserving its magnitude. The noise sequence $\hat{\boldsymbol{\epsilon}} = [\hat{\boldsymbol{\epsilon}}_1, \ldots, \hat{\boldsymbol{\epsilon}}_f]$ is denoised using a DDIM diffusion model $\mathcal{M}$. For each frame, DDIM iterates over $T$ steps:

$$
\begin{aligned}
\hat{\boldsymbol{z}}_i^{(t-1)} = \sqrt{\alpha_{t-1}} & \left( \frac{\hat{\boldsymbol{z}}_i^{(t)} - \sqrt{1 - \alpha_t}\, \boldsymbol{\epsilon}_\theta(\hat{\boldsymbol{z}}_i^{(t)}, t)}{\sqrt{\alpha_t}} \right) \\
& + \sqrt{1 - \alpha_{t-1}}\, \boldsymbol{\epsilon}_\theta(\hat{\boldsymbol{z}}_i^{(t)}, t),
\end{aligned}
\tag{3}
$$

with $\hat{\boldsymbol{z}}_i^{(T)} = \hat{\boldsymbol{\epsilon}}_i$, producing latent $\hat{\boldsymbol{z}}_i^{(0)}$. The VAE decoder $\mathcal{D}$ generates the watermarked video $V = [\mathcal{D}(\hat{\boldsymbol{z}}_1^{(0)}), \ldots, \mathcal{D}(\hat{\boldsymbol{z}}_f^{(0)})]$.

To adapt to videos of varying lengths and increase diversity, we generate an extended message list $M = [m_1, \ldots, m_L]$, where $L > f_{\max}$ and $f_{\max}$ is the maximum supported frame count. For each video, we sample a start index $p \sim \text{Uniform}(0, L - f)$, selecting messages $m_i' = M[p+i]$ for $i \in \{1, \ldots, f\}$. These are encoded via PRC to produce $\mathbf{c}_i$. The random start index ensures diverse initializations across videos, improving security and reducing detectable patterns, while supporting arbitrary video lengths. This frame-wise approach resists temporal and spatial attacks. The overall framework and detailed implementation are illustrated in Fig. 2 and Algorithm 1, respectively.

## 4.2 Watermark Extraction

In this section, we present our watermark extraction process, which consists of three key functions: decoding, detection, and recovery. This approach effectively handles various attacks that may disrupt the video structure.

### 4.2.1 Decoding Function

$\hat{m} = \mathcal{D}_{PRC}(V)$ extracts the embedded message from a watermarked video $V$ with $f$ frames. We first recover the approximate initial noise for each frame using the DDIM inverse process:

$$\tilde{\boldsymbol{\epsilon}}_i = \mathcal{M}^{-1}(V_i), \quad i \in \{1, \ldots, f\}. \tag{4}$$

Then, the message bit for each frame is decoded according to the sign pattern of the recovered noise:

$$\hat{m}_i = \text{PRC.Decode}(k, \text{sign}(\tilde{\boldsymbol{\epsilon}}_i)) \tag{5}$$

where the Decode function extracts the message bit encoded in the sign pattern using the PRC key $k$ (details of the PRC algorithm can be found in supplementary materials), matching the encoding process where $\hat{\boldsymbol{\epsilon}}_i = \mathbf{c}_i \cdot |\boldsymbol{\epsilon}_i|$ and $\mathbf{c}_i \in \{-1, 1\}^{c \times h \times w}$. The complete decoded sequence is returned as $\hat{m} = [\hat{m}_1, \ldots, \hat{m}_f]$.

### 4.2.2 Detection Function

$\{p, d\} = \text{Detect}(m, \hat{m})$ determines whether the decoded message $\hat{m}$ contains the watermark message $m$. We compute the **homology distance** between these sequences, where the cost of insertion, deletion, and replacement operations is 1. To assess statistical significance, we generate $N$ random sequences $\{r^1, r^2, \ldots, r^N\}$ and compute their homology distances with $\hat{m}$. The p-value is:

$$p = \text{rank}(d_{\text{hom}}(m, \hat{m}))/N \tag{6}$$

where rank is the position of $d_{\text{hom}}(m, \hat{m})$ among all distances. The detection result is $d = \mathbf{1}_{p < \tau}$ with threshold $\tau$. If the p-value is less than $\tau$, there is a watermark with $m$. The homology distance calculation is defined as:

$$d_{\text{hom}}(m, \hat{m}) = 2 \cdot \mathbb{E}_{k \sim \mathcal{U}\{1, n\}} \left[ \mathbb{I}_{\{m_k \neq \hat{m}_k\}} \right] - 1 \tag{7}$$

The transformation linearly scales distances to the range $[-1, 1]$, enhancing the sensitivity of the detection mechanism.

**Temporal Matching Function** $m' = \mathcal{T}(m, \hat{m})$ is applied to align and recover the message. We first identify the indices $I$ in the message sequence $m$ where the decoded message $\hat{m}$ occurs:

$$I_j = \arg\min_j \{d_{\text{hom}}(m, \hat{m}_j)\}, \quad j \in \{1, \ldots, f\} \tag{8}$$

Then, using the indices, we identify both the starting index $s$ and the optimal alignment path between $m$ and $\hat{m}$:

$$s, \mathcal{P} = \arg\min\{\{I_j\}, \text{Path}(m[i:], \hat{m})\} \tag{9}$$

where $\mathcal{P}$ represents the sequence of operations (match, insert, delete, substitute) that transforms $m[s:]$ into $\hat{m}$ with minimal cost. Using this path, we recover the original message by extracting the corresponding subsequence from $m$ that aligns with $\hat{m}$:

$$m' = \{m[s + j] \mid \mathcal{P}_j \text{ is a match or substitute operation}\} \tag{10}$$

This extracts precisely the elements from the original message that correspond to the decoded sequence after accounting for any frame manipulations.

## 5 Experiments

### 5.1 Experimental Setting

#### 5.1.1 Implementation details.

In our primary experiments, we explore both text-to-video (T2V) and image-to-video (I2V) generation tasks, employing ModelScope (MS) Wang et al. (2023) for T2V synthesis and I2VGen-XL Zhang et al. (2023a) for I2V generation. The generated videos consist of 16 frames, each with a resolution of $512 \times 512$. The inference and inversion steps are set to their default values of 25 and 50, respectively. Watermarks of 512 bits are embedded into each generated frame of the two models. As described in the Section 4.2, we leverage DDIM inversion to obtain predicted initial noise. The threshold $\tau$ is set to 0.005. The number of random sequences $N$ is set to 1000. All experiments are conducted on an NVIDIA Tesla A800 80G GPU.

#### 5.1.2 Baseline.

We selected four watermarking methods as baselines for comparison: RivaGANZhang et al. (2019), REVMarkZhang et al. (2023b), VideoSealFernandez et al. (2024), and VideoShieldHu et al. (2025). All selected methods are open-source and specifically designed to embed multi-bit strings within a video. Specifically, we set 32 bits for RivaGAN, 96 bits for REVMark, 96 bits for VideoSeal and 512 bits for VideoShield. Among these approaches, VideoShield stands out as the only in-generation method, whereas the others rely on post-processing techniques that necessitate training independent models for watermark embedding.

#### 5.1.3 Datasets.

We select 50 prompts from the test set of VBenchHuang et al. (2024), covering five categories: Animal, Human, Plant, Scenery, and Vehicles, with 10 prompts per category. For the T2V task, we generate four videos for each prompt for evaluation, ensuring diversity in outputs while maintaining consistency in prompt interpretation. For the T2V task, we first leverage a text-to-image model Stable Diffusion 2.1Rombach et al. (2022a), to generate images corresponding to the selected prompts. These generated images are subsequently utilized to create videos. Overall, we generate a total of 200 videos for both tasks for the primary experiments. Additionally, for each prompt category in VBench, we generate 10 watermarked and 10 non-watermarked videos, resulting in a total of 8,000 watermarked and 8,000 non-watermarked videos for the watermark learnability comparison experiment.

Table 1: Main results of *VideoMark*. All columns report bit accuracy, except for the Video Quality column.

| Model | Method | Extraction | | Video Quality | | | | | Temporal Tampering | | | | Spatial Tampering | | | |
|---|---|---|---|---|---|---|---|---|---|---|---|---|---|---|---|---|
| | | Bit Len. | Acc. | SC | BC | MS | IQ | Avg. | Swap | Insert | Drop | Avg. | G. Blur | C. Jitter | R. Comp. | Avg. |
| MS | RivaGAN | 32 | 0.994 | 0.922 | 0.951 | 0.960 | 0.648 | 0.870 | 0.930 | 0.919 | 0.930 | 0.926 | 0.919 | 0.939 | 0.783 | 0.880 |
| | REVMark | 96 | 0.996 | 0.943 | 0.960 | 0.972 | 0.450 | 0.831 | 0.992 | - | - | - | 0.987 | 0.765 | 0.508 | 0.753 |
| | VideoSeal | 96 | 0.964 | 0.950 | 0.959 | **0.977** | 0.679 | 0.891 | 0.960 | 0.960 | 0.961 | 0.961 | 0.964 | 0.964 | 0.565 | 0.831 |
| | VideoShield | 512 | **1.000** | 0.949 | **0.962** | **0.977** | 0.689 | 0.894 | **1.000** | - | - | - | **1.000** | **1.000** | 0.999 | 1.000 |
| | *VideoMark* | 512×16 | **1.000** | **0.951** | 0.961 | **0.977** | 0.692 | 0.895 | **1.000** | **1.000** | **1.000** | **1.000** | **1.000** | **1.000** | **1.000** | **1.000** |
| I2V | RivaGAN | 32 | 0.942 | 0.858 | 0.912 | 0.927 | 0.561 | 0.815 | 0.919 | 0.909 | 0.919 | 0.916 | 0.886 | 0.893 | 0.781 | 0.853 |
| | REVMark | 96 | 0.975 | 0.853 | 0.900 | 0.918 | 0.500 | 0.793 | 0.967 | - | - | - | 0.928 | 0.713 | 0.518 | 0.720 |
| | VideoSeal | 96 | 0.982 | 0.859 | 0.915 | 0.928 | 0.573 | 0.819 | 0.980 | 0.980 | 0.981 | 0.981 | 0.980 | **0.981** | 0.633 | 0.865 |
| | VideoShield | 512 | 0.990 | 0.811 | 0.892 | 0.913 | 0.530 | 0.787 | 0.990 | - | - | - | **0.990** | 0.849 | 0.777 | 0.872 |
| | *VideoMark* | 512×15 | **0.997** | **0.864** | **0.917** | **0.930** | **0.581** | **0.823** | **0.997** | **0.991** | **0.997** | **0.995** | 0.857 | 0.955 | **0.921** | **0.911** |

### 5.1.4 Metric.

We leverage Bit Accuracy to evaluate the ratio of correctly extracted watermark bits. To evaluate the quality of the generated videos, we conducted both objective and subjective assessments. For the objective evaluation, we leverage the metrics Subject Consistency, Background Consistency, Motion Smoothness, and Image Quality from VBench (see supplementary materials). For the subjective evaluation, we meticulously designed a pipeline that leverages GPT-4o to evaluate the generated videos (see supplementary materials).

## 5.2 Main Results

In Table 1, we present the main experimental results of VideoMark, including extraction accuracy, video quality, and both temporal and spatial robustness.

### 5.2.1 Extraction.

The "Extraction" columns present the watermark bit length and bit accuracy of VideoMark in comparison with the baseline methods. For I2V, due to the accumulation of significant errors in the first frame during the inversion stage, we embed the watermark in all frames except the first frame. VideoMark achieves bit accuracies of 1.000 and 0.997 on the two models while embedding 512×16 and 512×15 watermark bits, respectively, demonstrating superior extraction performance and confirming the effectiveness of our approach. This performance is comparable to the SOTA watermarking algorithm VideoShield, prominently surpassing other algorithms, including VideoSeal and REVMark.

### 5.2.2 Quality.

The "Video Quality" columns present the objective experimental results of various watermarking methods on the VBench benchmark. VideoMark consistently achieves state-of-the-art performance across all four metrics in both tasks. In the I2V task, it surpasses the best post-processing method, VideoSeal, by 0.004, and outperforms the leading in-processing method, VideoShield, by 0.036. Notably, in terms of Image Quality (IQ), our method achieves scores of 0.692 on T2V and 0.581 on I2V.

In addition to objective metrics, we employ an LLM-as-a-judge strategy alongside human voting for a comprehensive subjective quality assessment. For the LLM-based evaluation, we leverage GPT-4o to assess the perceptual quality of the generated content (the specific configuration is detailed in the preceding section). From the 8,000 videos generated by each model, we randomly sample 1,000 instances for evaluation. As shown in Table 2, VideoMark achieves the highest number of top-rated samples in both tasks—288 in T2V and 245 in I2V. This significantly outperforms the second-best methods, VideoShield and RivaGAN, by 48 and 33 samples, respectively. The visual results are provided in the supplementary material.

### 5.2.3 Robustness.

The "Temporal Tampering" and "Spatial Tampering" columns show robustness results under temporal and spatial attacks, respectively (detailed experimental settings are in supplementary materials).

Table 2: Quality Assessment via GPT-4o. We report the total number of top-rated samples assigned by GPT-4o across different base models. The methods compared include RivaGAN (RG), REVMark (RM), VideoSeal (VS), VideoShield (VSh), and *VideoMark* (*VM*).

| Model | RG | RM | VS | VSh | *VM* |
|---|---|---|---|---|---|
| ModelScope | 231 | 47 | 194 | 240 | **288** |
| I2VGen-XL | 222 | 111 | 205 | 217 | **245** |
| **Total** | 453 | 158 | 399 | 457 | **533** |

Table 3: Quantitative assessment of VideoMark resistance to temporal tampering via (*p*-values).

| ModelScope | | | I2VGen-XL | | |
|---|---|---|---|---|---|
| Swap | Insert | Drop | Swap | Insert | Drop |
| 0.001 | 0.001 | 0.001 | 0.001 | 0.001 | 0.001 |

Table 4: Comparison of temporal robustness, using matching accuracy as the evaluation metric.

| Method | ModelScope | | | I2VGen-XL | | |
|---|---|---|---|---|---|---|
| | Swap | Insert | Drop | Swap | Insert | Drop |
| VideoShield | 1.000 | 1.000 | 1.000 | 0.983 | 0.983 | 0.981 |
| *VideoMark* | 1.000 | 1.000 | 1.000 | **0.996** | **0.989** | **0.996** |

For temporal tampering, we show the bit accuracy between the decoded and embedded message. As REVMark does not release the necessary model files, and VideoShield cannot handle videos with variable frames during decoding, we omit their results from this evaluation. The results show that VideoMark maintains a perfect bit accuracy of 1.000 in the T2V task. In the I2V task, it achieves an average bit accuracy of 0.996 and retains a strong performance of 0.991 even under the most challenging frame insertion attack. These findings further demonstrate that VideoMark can reliably recover the embedded message even in the presence of temporal tampering, indicating that the watermark is effectively and redundantly distributed across frames to ensure robust decoding.

In addition, in Table 3, we present the p-values of VideoMark's detection results under temporal tampering attacks. Both models exhibit a p-value of 0.001 in detecting temporal tampering, which indicates strong statistical significance. Table 4 compares frame matching accuracy between VideoMark and VideoShield. VideoMark achieves an accuracy of 0.996 on the I2V benchmark, empirically confirming the efficacy of the proposed TMM module in preserving and reconstructing high-fidelity temporal sequences.

For spatial tampering (details in supplementary materials), VideoMark embeds 32 bits per frame, achieving perfect bit accuracy (1.000) on the T2V task. In the I2V task, despite a lower score under Gaussian Blur (0.857), it attains the highest average accuracy (0.911) across all attacks.

### 5.2.4 Invisibility.

To evaluate detectability, we leverage VideoMAE Tong et al. (2022) as the backbone and train it with 100 epochs on a dataset consisting of 8,000 watermark-free videos and 8,000 watermarked videos to perform binary classification. The results in Figure 3, show that the network's classification accuracy is notably low for videos watermarked with VideoMark, achieving 54.07% on the training set and 48.02% on the validation set. In contrast, other watermarking methods show similar performance on training and validation sets, indicating their watermark patterns are easier to learn and detect.

### 5.3 Analysis

### 5.3.1 Impact of video length.

To comprehensively evaluate how the number of generated frames affects watermarking performance, we report both extraction accuracy and visual quality across different generation lengths in Fig. 4. We observe that the two metrics degrade to different extents as the number of frames increases from 16 to 32. Extraction accuracy drops from 1.000 to 0.925 in the T2V task and from 0.997 to 0.816 in the I2V task, which indicates that the root cause lies in the increasing difficulty of accurately recovering the original noise as the number

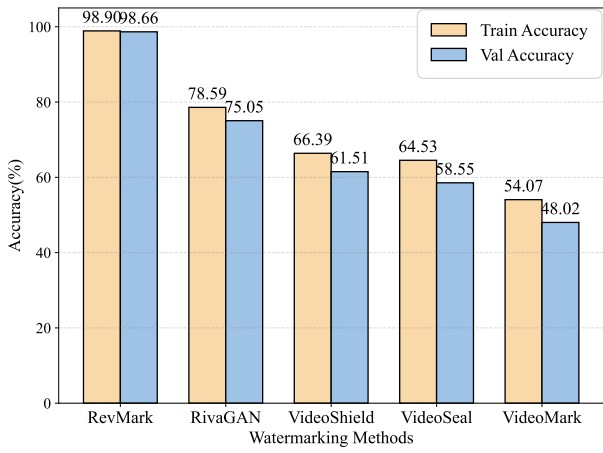

Figure 3: The binary classification results under different watermarking algorithms.

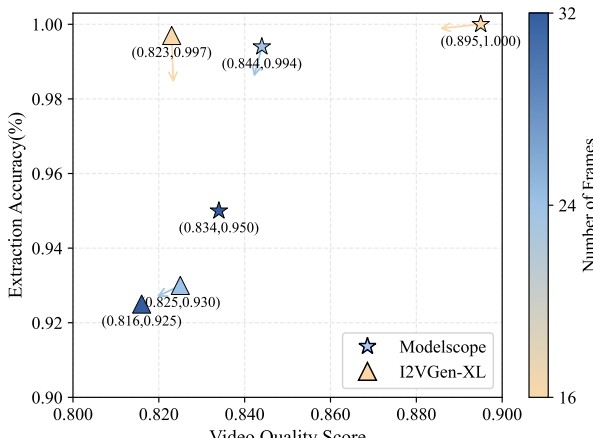

Figure 4: The extraction accuracy and visual quality score of the watermarking scheme across different models and frame counts.

Table 5: Evaluation of bit accuracy across varying spatial resolutions.

| Model | Spatial Resolution | | | |
|---|---|---|---|---|
| | $256 \times 256$ | $512 \times 512$ | $960 \times 544$ | $1280 \times 720$ |
| Modelscope | 1.000 | **1.000** | 1.000 | 0.998 |
| I2VGen-XL | 0.996 | **0.997** | 0.980 | 0.973 |

of frames rises. This leads to larger cumulative errors and, consequently, a decline in extraction accuracy. Meanwhile, to explore the impact of VideoMark on visual quality, we compare the distribution of video quality scores between watermarked and non-watermarked videos in Fig. 5. The distribution of video quality scores for watermarked videos remains consistent with that of clean videos across different frame lengths, which indicates that VideoMark introduces minimal perceptual distortion, regardless of the video length. The observed degradation in video fidelity is primarily attributable to the inherent architectural constraints in modeling long-range temporal dependencies, which limit the model's generative capacity for extended sequences.

### 5.3.2 Impact of message length.

As shown in Fig.6, we evaluate the extraction capability and robustness across different message lengths. In both tasks, the extraction accuracy remains stable when the message length is below 512, but drops for longer messages, since the redundant bits have reached their limit in providing effective error correction. It indicates that VideoMark can stably embed up to 512 bits of messages and extract them reliably in the absence of attacks. Based on these findings, we fix the message length at 512 bits for all subsequent extraction experiments. Additionally, we evaluate the robustness of VideoMark against three types of spatial tampering. Robustness in the T2V task peaks at 32 watermark bits but degrades as bits increase. Conversely, I2V robustness peaks at 64 bits, slightly surpassing the 32-bit case. We attribute these differences to the higher generative complexity of T2V models, whose greater output variability likely induces more spatial perturbations, making robustness more sensitive to embedding payload.

### 5.3.3 Impact of spatial resolution.

To evaluate the extraction capability of the watermark at different resolutions, we present the watermark bit accuracy across various resolutions in Table 5. In the T2V task, the extraction accuracy remains at 0.998 even at a resolution of $1280 \times 720$, demonstrating strong extraction performance. In contrast, in the

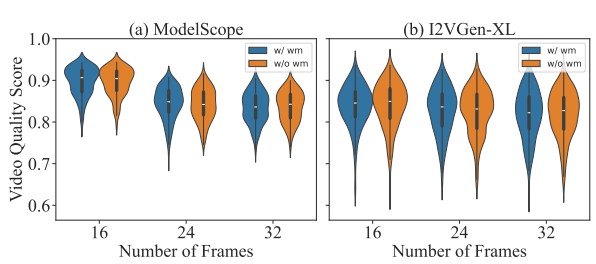

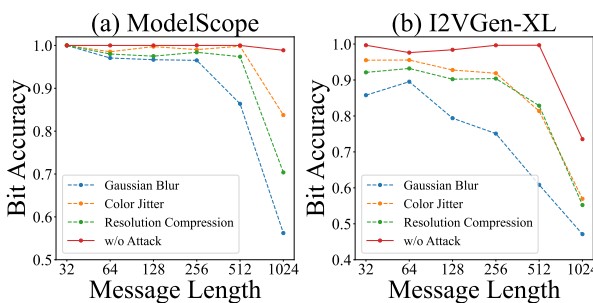

Figure 5: Video quality scores under varying numbers of frames, comparing watermarked (w/ wm) and clean (w/o wm) outputs.

Figure 6: The extraction accuracy and robustness of VideoMark against spatial tampering for varying message lengths.

I2V task, extraction accuracy peaks at a resolution of $512 \times 512$. We attribute this to a balanced trade-off between VideoMark's error correction capability and inversion errors at this resolution. In contrast, higher resolutions introduce larger inversion errors during the inversion process, which hinder watermark extraction.

### 5.3.4 Impact of the inversion step.

To evaluate the impact of mismatch steps between inference and inversion, we present the extraction accuracy at different steps in Table 6. In the T2V task, mismatched steps introduce minimal loss in extraction accuracy, while in the I2V task, they lead to a significant accuracy degradation. We attribute this to the fact that T2V models are typically more robust to small variations in the inversion process, while I2V models are more sensitive to such discrepancies, leading to greater performance degradation. Considering practical implementation efficiency and extraction capability, we fix the inference and inversion steps at 25 for the T2V and 50 for the I2V.

Table 6: Bit accuracy across different inference (Inf.) and inversion (Inv.) steps. The main experimental configuration is marked.

| Inv. Inf. | ModelScope | | | | I2VGen-XL | | | |
|---|---|---|---|---|---|---|---|---|
| | 10 | 25 | 50 | 100 | 10 | 25 | 50 | 100 |
| 10 | 0.986 | 0.995 | 0.991 | 1.000 | 0.922 | 0.903 | 0.892 | 0.915 |
| 25 | 0.998 | **1.000** | 1.000 | 1.000 | 0.940 | 0.985 | 0.982 | 0.989 |
| 50 | 0.999 | 1.000 | 0.999 | 1.000 | 0.957 | 0.981 | **0.997** | 0.993 |
| 100 | 1.000 | 1.000 | 1.000 | 1.000 | 0.957 | 0.987 | 0.988 | 0.993 |

## 6 Conclusion

In this work, we propose a training-free, undetectable watermarking framework for video diffusion models. Through extensive experiments, we demonstrate that the generated videos retain high visual quality and exhibit no perceptible artifacts attributable to the embedded watermark. However, the current framework relies on approximate inversion techniques, which limit extraction accuracy in certain scenarios. For future improvements, we suggest exploring more advanced or robust inversion algorithms to enhance the reliability and effectiveness of the watermark retrieval process in real-world AIGC applications.

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

## A  Appendix

### A.1  PRC Watermark

#### A.1.1  Generation phase.

To ensure the imperceptibility of the embedded watermark, we define a stochastic modulation process that preserves the statistical manifold of the latent space. Let $(\Omega, \mathcal{F}, \mathbb{P})$ be the underlying probability space. We introduce an independent Rademacher random variable $C : \Omega \to \{-1, 1\}$ such that $\mathbb{P}(C = 1) = \mathbb{P}(C = -1) = 0.5$, and a standard Gaussian noise vector $\epsilon \sim \mathcal{N}(0, \mathbf{I})$.

The watermarking transformation is defined as $\hat{\epsilon} = \Phi(\epsilon, C) = |\epsilon| \cdot C$. To prove that $\hat{\epsilon}$ is distributionally indistinguishable from the original noise $\epsilon$, we examine its **characteristic function** $\varphi_{\hat{\epsilon}}(t)$:

$$
\begin{aligned}
\varphi_{\hat{\epsilon}}(t) &= \mathbb{E}\left[e^{it\hat{\epsilon}}\right] = \mathbb{E}_C\left[\mathbb{E}_\epsilon\left[e^{itC|\epsilon|} \mid C\right]\right] \\
&= \mathbb{P}(C = 1)\mathbb{E}\left[e^{it|\epsilon|}\right] + \mathbb{P}(C = -1)\mathbb{E}\left[e^{-it|\epsilon|}\right] \\
&= \frac{1}{2}\int_0^\infty \sqrt{\frac{2}{\pi}}e^{-\frac{u^2}{2}}\left(e^{itu} + e^{-itu}\right)du
\end{aligned}
\tag{11}
$$

By invoking Euler's formula, the integrand simplifies to the cosine kernel:

$$
\varphi_{\hat{\epsilon}}(t) = \int_0^\infty \sqrt{\frac{2}{\pi}}e^{-\frac{u^2}{2}}\cos(tu)du = e^{-\frac{t^2}{2}}
\tag{12}
$$

The resulting expression $e^{-\frac{t^2}{2}}$ is the unique characteristic function of the standard normal distribution $\mathcal{N}(0, 1)$. According to the **Lévy Continuity Theorem**, we have $\hat{\epsilon} \overset{d}{=} \epsilon$.

This distributional isomorphism guarantees that the watermarked latents reside strictly within the original Gaussian prior. Consequently, the modification remains theoretically undetectable by any statistical adversary, achieving perfect steganographic transparency while maintaining the high-fidelity generation capabilities of the diffusion backbone.

#### A.1.2  Detection phase.

**PRC.Detect** accepts as input a vector $\mathbf{s}$ with entries in the interval $[-1, 1]$ rather than a bit-string. Given the PRC codeword $\mathbf{c}$, each component $s_i$ is intended to approximate the conditional expectation of $(-1)^{c_i}$ based on the user's observation. The full procedure for **PRC.Detect** is presented in Algorithm 2.

---

**Algorithm 2:** PRC.Detect

**Input**  : PRC-key $k$, $s$
**Output:** Detection result True or False

Parse parameters from key $k$: $(n, \text{msg\_len}, F, t, \lambda, \eta, \text{bits}, k, r, \text{iter}, \text{otp}, \text{testbits}, G, P) \leftarrow k$;
**for** $i \in [n]$ **do**
  $\quad s_i \leftarrow (-1)^{\text{otp}_i} \cdot (1 - 2\eta) \cdot s_i$;
**for** *each parity check $w \in P$* **do**
  $\quad$ Let $\hat{s}_w \leftarrow \prod_{i \in w} s_i$;
$C \leftarrow \frac{1}{2}\sum_{w \in P} \log^2\left(\frac{1+\hat{s}_w}{1-\hat{s}_w}\right)$;
**if** $\sum_{w \in P} \log\left(\frac{1+\hat{s}_w}{2}\right) \geq \sqrt{C\log(1/F)} + \frac{1}{2}\sum_{w \in P}\log\left(\frac{1-\hat{s}_w^2}{4}\right)$ **then**
  $\quad$ **return** True;
**else**
  $\quad$ **return** False;

---

### A.1.3 Decode phase.

**PRC.Decode** accepts as input a vector **s** with entries in the interval $[-1, 1]$, and returns a decoded binary message. The full procedure for **PRC.Decode** is presented in Algorithm 2.

---

**Algorithm 3:** PRC.Decode

---

**Input** : PRC-key $k$, $s$
**Output:** Decoded message $m \in \{0, 1\}^k$ or None

Parse parameters from key $k$: $(n, \text{msg\_len}, F, t, \lambda, \eta, \text{bits}, k, r, \text{iter}, \text{otp}, \text{testbits}, G, P) \leftarrow k$;
**for** $i \in [n]$ **do**
$\quad \lfloor \ s_i \leftarrow (-1)^{\text{otp}_i} \cdot (1 - 2\eta) \cdot s_i$;
$y \leftarrow \text{BP-OSD}(G, P, s)$;
Parse $(\text{testbits}, r, m) \leftarrow y$;
**return** $m$;

---

Algorithm 2 performs coarse-grained detection on each video frame to preliminarily identify potential targets or anomalies, serving as a foundation for the overall detection of the video.

In contrast, Algorithm 3 provides fine-grained analysis by attempting to decode a binary message from the input signal, enabling precise verification of target presence and integrity.

Table 7: Video quality comparison across watermarking methods. VideoMark achieves the highest or near-highest scores across all metrics, closely matching clean videos.

| Model | Method | Subject Consistency | Background Consistency | Motion Smoothness | Imaging Quality |
|---|---|---|---|---|---|
| MS | RivaGAN | 0.922 | 0.951 | 0.960 | 0.648 |
| | REVMark | 0.943 | 0.960 | 0.972 | 0.450 |
| | VideoSeal | 0.950 | 0.959 | **0.977** | 0.679 |
| | VideoShield | 0.949 | **0.962** | **0.977** | 0.689 |
| | *VideoMark (ours)* | **0.951** | 0.961 | **0.977** | **0.692** |
| | w/o watermark | **0.951** | **0.962** | **0.977** | 0.691 |
| I2V | RivaGAN | 0.858 | 0.912 | 0.927 | 0.561 |
| | REVMark | 0.853 | 0.900 | 0.918 | 0.500 |
| | VideoSeal | 0.859 | 0.915 | 0.928 | 0.573 |
| | VideoShield | 0.811 | 0.892 | 0.913 | 0.530 |
| | *VideoMark (ours)* | **0.864** | **0.917** | **0.930** | **0.581** |
| | w/o watermark | 0.861 | 0.913 | 0.928 | 0.578 |

Table 8: Detailed performance evaluation of videos assessed by GPT-4o. Metrics include SC (Subject Consistency), BC (Background Consistency), MS (Motion Smoothness), and IQ (Imaging Quality).

| Method | ModelScope | | | | Total | I2VGen-XL | | | | Total |
|---|---|---|---|---|---|---|---|---|---|---|
| | SC | BC | MS | IQ | | SC | BC | MS | IQ | |
| RivaGAN | **7.88** | 8.85 | **4.75** | 6.90 | **28.38** | **7.69** | 7.67 | **6.32** | 7.14 | **28.82** |
| REVMark | 6.52 | 7.95 | 4.41 | 3.95 | 22.83 | 7.62 | **7.73** | 5.94 | 5.57 | 26.86 |
| VideoSeal | 7.82 | 8.86 | 4.65 | 6.90 | 28.23 | 7.63 | 7.62 | 6.19 | 7.10 | 28.54 |
| VideoShield | 7.77 | 8.86 | 4.37 | 6.85 | 27.85 | 7.58 | 7.57 | 6.17 | 7.04 | 28.36 |
| *VideoMark (ours)* | 7.86 | **8.93** | 4.55 | **6.98** | 28.32 | 7.66 | 7.70 | 6.10 | **7.19** | 28.65 |
| w/o watermark | 7.84 | 8.84 | 4.67 | 6.95 | 28.31 | 7.67 | 7.61 | 6.28 | 7.17 | 28.73 |

### A.2 VBench for Video Quality

This subsection provides an overview of VBench video quality metrics—Subject Consistency, Background Consistency, Motion Smoothness, and Imaging Quality—with further details available in the original paper.

- **Subject Consistency** is measured by computing the similarity of DINO featuresCaron et al. (2021) across frames:

$$S_{\text{subject}} = \frac{1}{T-1} \sum_{t=2}^{T} \frac{1}{2} \left( \langle d_1, d_t \rangle + \langle d_{t-1}, d_t \rangle \right), \tag{13}$$

where $d_i$ denotes the DINO image feature of the $i^{\text{th}}$ frame, normalized to unit length, and $\langle \cdot, \cdot \rangle$ represents the dot product, corresponding to cosine similarity.

- **Background Consistency** assesses the temporal coherence of background scenes by computing CLIP feature similarity across frames Radford et al. (2021):

$$S_{\text{background}} = \frac{1}{T-1} \sum_{t=2}^{T} \frac{1}{2} \left( \langle c_1, c_t \rangle + \langle c_{t-1}, c_t \rangle \right), \tag{14}$$

where $c_i$ denotes the CLIP image feature of the $i^{\text{th}}$ frame, normalized to unit length.

- **Motion Smoothness** is assessed using a frame-by-frame motion prior inspired by video frame interpolation models Li et al. (2023). Given a generated video with frames $[f_0, f_1, f_2, f_3, f_4, \ldots, f_{2n-2}, f_{2n-1}, f_{2n}]$, the odd-indexed frames are manually removed to create a low-frame-rate sequence $[f_0, f_2, f_4, \ldots, f_{2n-2}, f_{2n}]$. A video frame interpolation model Li et al. (2023) is then applied to reconstruct the dropped frames $[\hat{f}_1, \hat{f}_3, \ldots, \hat{f}_{2n-1}]$. The Mean Absolute Error (MAE) between the reconstructed and original frames is calculated and normalized to the range $[0, 1]$, where a higher value indicates smoother motion.

- **Imaging Quality** is evaluated using the MUSIQ image quality predictor Ke et al. (2021), which is trained on the SPAQ dataset Fang et al. (2020) and is capable of handling images with varying aspect ratios and resolutions. Each frame is assigned a quality score by MUSIQ, which is linearly normalized to the range $[0, 1]$ by dividing by 100. The final imaging quality score is computed by averaging the normalized scores across all frames in the video.

In Table 7, we present a comparison between the video quality with and without watermarking under the VBench benchmark. It can be observed that VideoMark preserves video quality comparable to that of watermark-free videos.

### A.3   MLLM-as-a-Judge for Video Quality

Figure 7 shows the specific evaluation prompt used with GPT-4o for video quality assessment. The specific results are presented in Table 8, where each evaluation metric demonstrates that the proposed Videomark method effectively preserves the visual quality of the videos while embedding the watermark. Although there is a slight gap in subject consistency and motion smoothness compared to RivaGAN, the overall performance is comparable to that of the other methods.

### A.4   Temporal Tampering Implement

We provide the details of the temporal tampering in this subsection. We consider three types of temporal tampering: **Frame Drop**, **Frame Insert**, and **Frame Swap**. The first two tampering methods alter the total number of frames in the video, whereas the last method only alters the frame order. The implementations and formulations are described as follows. Given a video sequence with $f$ frames:

- **Frame Drop** removes a frame at a randomly selected temporal index $p$, where $p \in \{0, 1, \ldots, f-1\}$. The tampered sequence is defined as:

$$\text{Drop}(\text{frame}_p) \rightarrow \{\text{frame}_0, \ldots, \text{frame}_{p-1},$$
$$\text{frame}_{p+1}, \ldots, \text{frame}_{f-1}\}.$$

---

**Prompts for GPT-4o Scoring Video Quality**

Please perform a thorough and objective evaluation of the video result for sample 'sample_name'. Using the provided 16 sampled frames as a reference, carefully assess the video on the following criteria:

**1. Subject Consistency:** Evaluate how consistently and accurately the subject is represented across all frames.
**2. Background Consistency:** Assess the uniformity and coherence of the background elements throughout the video.
**3. Motion Smoothness:** Judge the fluidity and natural progression of motion, ensuring transitions are smooth and realistic.
**4. Imaging Quality:** Evaluate the overall visual quality, including clarity, sharpness, color fidelity, and detail.

For each criterion, assign a numeric score between 0 (lowest) and 10 (highest) based solely on the visual data. Compute the total score as the sum of these individual scores. Your evaluation should be objective, data-driven, and free from any additional commentary or explanation. Return your response strictly as a JSON object formatted exactly as shown below without any extra text:

```
{
"subject_consistency":  <number>,
"background_consistency":  <number>,
"motion_smoothness":  <number>,
"imaging_quality":  <number>,
"total_score":  <number>
}
```

Figure 7: Prompt used for the subjective evaluation of video quality.

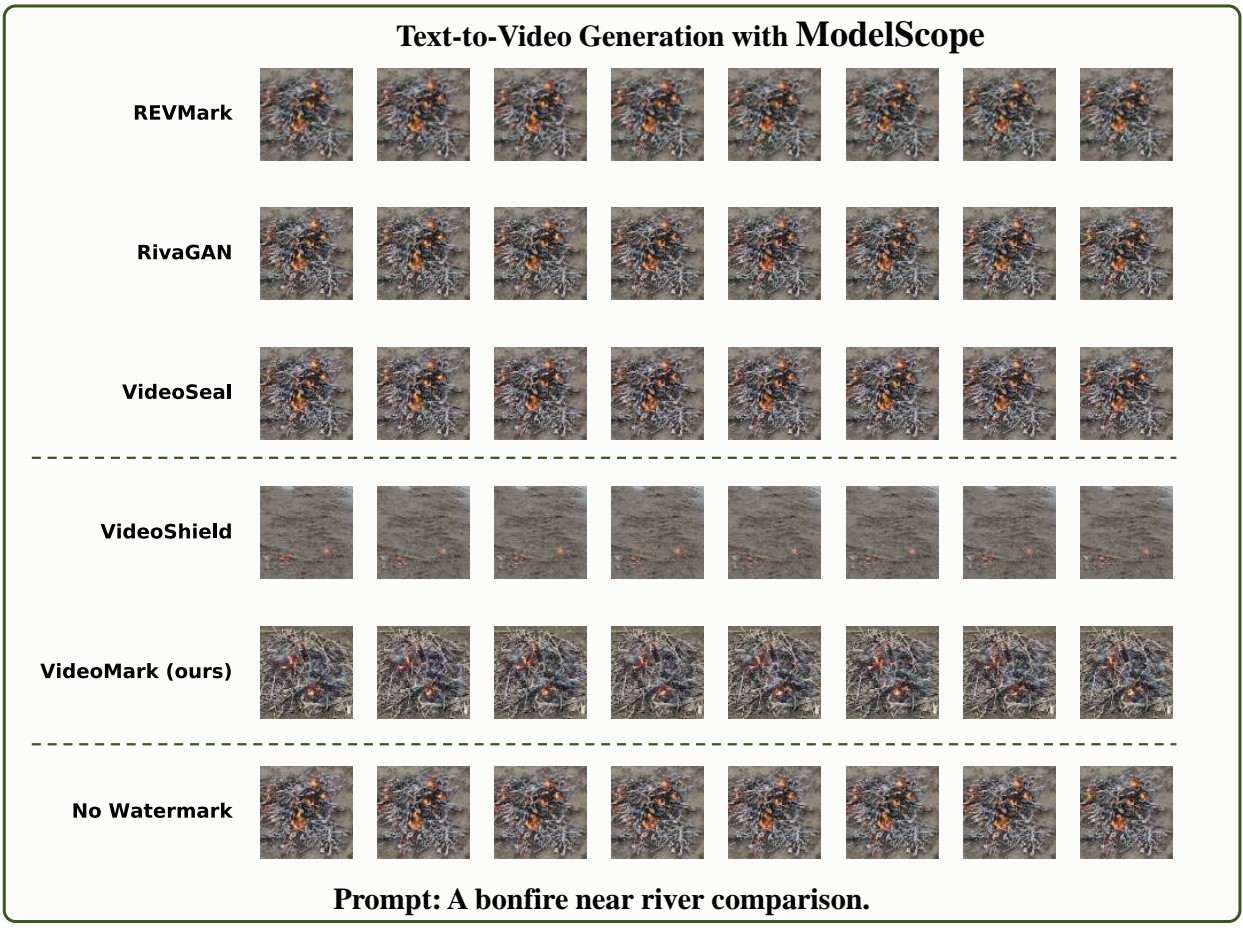

Figure 8: Visualization results comparing VideoMark with existing approaches on the T2V task.

- **Frame Insert** duplicates a neighboring frame and inserts it at a randomly selected index $p \in \{1, 2, \ldots, f - 1\}$. Assuming the duplicated frame is the preceding frame, the tampered sequence becomes:

$$\text{Insert}(\text{frame}_{p-1}) \rightarrow \{\text{frame}_0, \ldots, \text{frame}_{p-1},$$
$$\text{frame}_{p-1}, \text{frame}_p, \ldots, \text{frame}_{f-1}\}.$$

- **Frame Swap** exchanges the positions of temporally adjacent frames at regular intervals. Specifically, for

$$p = 2 + 4k, \quad k \sim \left\{0, 1, 2, \ldots, \left\lfloor \frac{f - 3}{4} \right\rfloor \right\},$$

the swap operation is defined as:

$$\text{Swap}(\text{frame}_p, \text{frame}_{p+1}) \rightarrow \{\ldots, \text{frame}_{p+1}, \text{frame}_p, \ldots\}.$$

## A.5 Spatial Tampering Implement

We provide the details of the spatial tampering in this subsection. We consider three types of temporal tampering: **Gaussian Blur**, **Colour Jitter**, **Resolution Compression**. These tampering operations are applied to every frame of the video. The implementations and formulations are described as follows. Given a video frame $\mathbf{F}$ with height $H$ and width $W$:

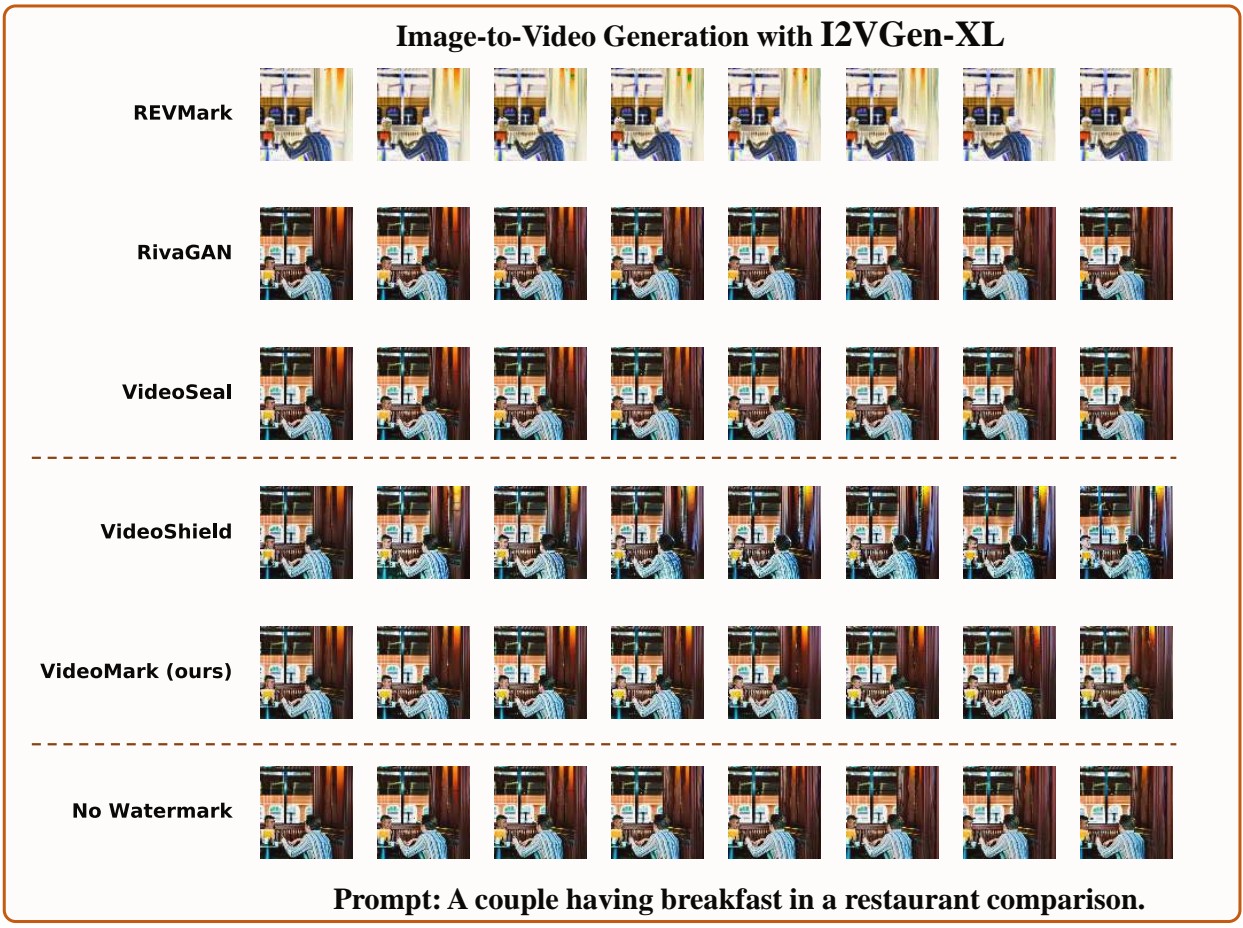

Figure 9: Visualization results comparing VideoMark with existing approaches on the I2V task.

- **Gaussian Blur** applies a convolution with a Gaussian kernel $K$ of size $k \times k$, where $k$ is a positive odd integer. The blurred frame $\mathbf{F}'$ is computed as:

$$\mathbf{F}' = \mathbf{F} * K,$$

  where $*$ denotes the convolution operation. We use a kernel size of $11 \times 11$ in our experiments.

- **Colour Jitter** perturbs the brightness and contrast of the frame. The transformed frame $\mathbf{F}'$ is computed as:

$$\mathbf{F}' = \alpha \mathbf{F} + \beta,$$

  where $\alpha \sim \mathcal{U}(1 - s, 1 + s)$ controls contrast, $\beta \sim \mathcal{U}(-s, s)$ controls brightness, and $s$ is the jitter strength hyperparameter. We set $s = 0.1$ in all experiments.

- **Resolution Compression** simulates compression artifacts by downscaling and then upscaling the frame. Given a downscale factor $r \in (0, 1)$, the frame $\mathbf{F}'$ is obtained as:

$$\mathbf{F}' = \text{Upscale}(\text{Downscale}(\mathbf{F}, r), (H, W)),$$

  where $\text{Downscale}(\cdot, r)$ resizes the frame to dimensions $(rH, rW)$ and $\text{Upscale}(\cdot, (H, W))$ resizes it back to the original size using bilinear interpolation. We set a downscale factor of $r = 0.5$.

**A.6    Visualization of Generated Videos**

This subsection presents qualitative visualizations of generated videos for both T2V and I2V tasks. We show the first eight frames of each video to compare post-processing methods, in-generation approaches, and watermark-free baselines.

For the T2V task (Figure 8), our method produces videos that closely resemble watermark-free outputs under identical prompts, outperforming VideoShield. Unlike post-processing approaches, our framework maintains content diversity without requiring additional watermark overlays.

In the I2V task (Figure 9), our method achieves superior temporal coherence and better subject color consistency compared to VideoShield. Unlike REVMark, which degrades frames significantly, our approach preserves frame integrity without noticeable distortion.

Overall, our method consistently surpasses existing watermarking techniques in both tasks by producing videos that retain visual fidelity, diversity, and temporal consistency, while avoiding frame degradation. These qualitative results highlight the practical effectiveness of our watermarking framework in real-world video generation.

