# OpenReview forum: "VideoMark: A Distortion-Free Robust Watermarking Framework for Video Diffusion Models"
_TMLR — Under review for TMLR_

### Review · Reviewer_S6iP · 2026-06-23

**Summary Of Contributions:**

The paper proposes a training-free, in-generation video watermarking framework, so-called "VideoMark," tailored for video diffusion models. The method is to address the weakness of the existing video watermarking methods (e.g., temporal attacks) while maintaining the visual quality of the generated video. Specifically, VideoMark incorporates pseudo-random error correction (PRC) codes to independently embed the extended message sequence into the initial noise of each frame. To resist temporal tampering in videos, the authors introduce a temporal matching module (TMM) based on coherence distance to align the decoded message with the original sequence. VideoMark achieves improvements in extraction accuracy, stronger invisibility, and superior visual quality compared to VideoShield.

Strengths:
1. VideoMark directly manipulates the initial noise through PRC, avoiding the computational overhead of training separate post-processing encoder/decoder networks.

Weakness:
1. The article suffers from rough writing, typographical errors, and inconsistent formatting. For example, the heading styles are not consistent throughout the paper: Sec. 4 uses title case without periods (e.g., "4.2.1 Decoding Function"), whereas Sec. 5 and the Appendix adopt sentence case with periods (e.g., "5.1.1 Implementation details."). The appendix also contains apparent copy-and-paste errors. For example, Sec. A.1.3 refers to "Algorithm 2", while Algorithm 3 is actually displayed.

2. The paper claims to utilize "soft decision" to optimize robustness. However, Eq. (5) strictly applies hard symbolic operations (i.e., sign functions) before decoding, completely violating the continuous confidence values ​​required for soft decision decoding.

3. There is serious misuse of mathematical notation in Sec. 4.2.2. For example, Eq. (9) uses the argmin operator but fails to define the actual objective/cost function. Eq. (7) defines the cohomology distance as a normalized Hamming distance, which mathematically fails to handle the textually claimed "insertion and deletion" costs.

4. In Sec. 5.2.1, the authors claim that they omitted the first frame of the I2V task "due to the accumulation of serious errors." However, no experimental evidence or ablation studies are provided to support this claim.

5. The robustness test results in Table 1 appear to have been obtained under conditions drastically different from the payload settings advertised by VideoMark. While the table lists capacities of "512×16" and "512×15" bits, Sec 5.2.3 notes that only 32 bits were embedded per frame in the spatial tampering experiment, and Sec 5.3.2 further states that robustness decreases as the payload size increases. Therefore, the reported near-perfect robustness results may not reflect the performance of the full-capacity configuration shown in Table 1. Such comparisons may overestimate the actual robustness of the proposed framework without evaluation under matching payload settings.

6. The robustness evaluation is substantially less rigorous than that of the main baseline, VideoShield. While the proposed method is tested only under a small number of spatial perturbations (e.g., resizing and blurring), VideoShield is evaluated against 11 diverse distortions, including H.264/MPEG-4 compression. Since video compression is one of the most common transformations encountered in practical deployment and is known to severely affect noise-based watermarks, excluding it from the evaluation raises concerns about the validity of the reported robustness results.

7. The MLLM-as-a-Judge evaluation in Appendix A.3 asks GPT-4o to assess motion smoothness from a static grid of 16 sampled frames. Since motion smoothness is inherently a temporal property, it is unclear how such an image-only protocol can reliably measure the claimed attribute. This raises concerns about the validity of the evaluation.

8. Based on Algorithm 3, the PRC.Decode function relies on the BP-OSD algorithm. VideoMark requires executing this complex decoding process independently on every single frame before the Temporal Matching Module (TMM) can even be applied. This makes it expensive in terms of computational costs.

**Additional Comments:**

While this paper explores an important question and contains some interesting perspectives, I remain reserved about the current experimental validation and certain methodological formulations. In particular, the fairness of robustness comparisons, the validity of subjective evaluation schemes, and some mathematical inconsistencies significantly weaken the paper's core arguments. I look forward to a revised version that addresses these issues.

**Audience:**

Yes

**Audience Explanation:**

**Yes**, the problem addressed in this paper is highly relevant and timely for the TMLR audience. As video diffusion models (e.g., Sora, ModelScope, SVD) continue to advance rapidly, concerns regarding copyright infringement, deepfakes, and content attribution have become urgent.

Specifically, the authors explore a significant and unique challenge in the field of video watermarking: vulnerability to temporal attacks (e.g., frame dropping, insertion, and swapping). As the authors point out, existing generative internal watermarking methods (i.e., VideoShield) typically rely on fixed-length latent initializations and are prone to dimensionality collapse when the temporal structure changes. The concept of maintaining temporal robustness without incurring additional training overhead using extended message sequences and temporal matching modules (TMMs) is conceptually very attractive.

Therefore, despite the serious deficiencies in current implementations, mathematical formulations, and experimental evidence (see above), the research questions explored and the exploration of watermarking mechanisms against temporal attacks are undoubtedly of great significance to researchers working on generative AI security, trusted AI, and diffusion models.

**Broader Impact Concerns:**

I do not identify any significant ethical concerns beyond those already discussed in the paper. The proposed method is primarily intended to improve content provenance and ownership protection for AI-generated videos, which may have positive societal benefits. No additional Broader Impact discussion appears necessary.

**Claims And Evidence:**

No

**Claims Explanation:**

**No**. While the paper presents extensive experiments and reports favorable results for VideoMark, the evidence is **not sufficiently convincing to support** several of the paper's central claims.

First, the **robustness evaluation** is substantially **less comprehensive than** that of the main baseline, VideoShield. Important real-world distortions, particularly **H.264/MPEG-4 compression**, are omitted despite being known to significantly affect watermark reliability. As a result, the reported robustness results **do not provide sufficient evidence** for practical deployment scenarios.

Second, some methodological choices are insufficiently justified. For example, the omission of the first frame in the I2V setting is attributed to "serious error accumulation," yet **no experimental analysis or ablation study is provided to validate this explanation**.

Third, the subjective evaluation protocol raises concerns regarding validity. The MLLM-as-a-Judge evaluation asks GPT-4o **to assess motion smoothness from a static montage of sampled frames**, even though motion smoothness is inherently a temporal property. It is therefore **unclear whether the reported scores actually measure the claimed attribute**.

Finally, several technical descriptions are **unclear or inconsistent with the stated methodology**. Examples include the claimed use of soft-decision decoding despite hard thresholding operations in Eq. (5), ambiguities in the formulation of Sec. 4.2.2, and inconsistencies between some reported results and the conclusions drawn in the main text.

Overall, these issues **weaken the connection between the experimental evidence and the paper's main claims**, making the overall support insufficiently convincing.

**Requested Changes:**

1. **Mathematical Consistency and Correctness.** Several formulas in Sec. 4.2 appear inconsistent with the textual descriptions or mathematically incomplete. For example, the optimization objective is not defined in formula (9); the insertion/deletion handling claimed in formula (7) does not match the distance formula; the symbols in formula (8) are ambiguous; and there is a clear contradiction between the soft-decision decoding claimed in formula (5) and the hard-thresholding operation. These issues require clarification and correction.

2. **Fair and Consistent Evaluation.** The robustness comparisons with VideoShield should be conducted under matched payload settings and evaluated against standard video distortion (including H.264/HEVC compression). Robustness curves at different distortion intensities should also be provided.

3. **Validity of Subjective Evaluation.** The motion smoothness evaluations based on GPT-4o should be revised or replaced, employing a scheme that directly evaluates temporal consistency. Furthermore, the conclusions in Table 8 should be consistent with the reported results.

4. **Payload Consistency.** The robustness evaluation should be performed using payload settings consistent with those reported in Table 1. In particular, spatial tampering robustness should be evaluated using the advertised 512-bit payload rather than a reduced 32-bit configuration. If robustness degrades at higher payloads, the corresponding results should be reported explicitly. At a minimum, Table 1 should clearly distinguish between the claimed watermark capacity and the payload used in robustness testing.

5. **Practicality Analysis.** This paper should provide runtime and computational overhead measurements for embedding and extraction, especially taking into account the dependencies on DDIM inversion and frame-by-frame BP-OSD decoding.

---

### Review · Reviewer_47aV · 2026-06-24

**Summary Of Contributions:**

- The paper proposes a reasonable watermarking approach for video diffusion models with sound design rationale.
- Although the experiments are limited in scope, the proposed method generally outperforms existing approaches.
- The paper itself is well written.

**Additional Comments:**

As strengths, I would like to mention the following:

- This paper is well written. The problems it identifies are clearly articulated, and each component of the proposed method addresses them in an interpretable manner. While the use of PRC does not appear to be strictly necessary, it seems like a clever design choice. In particular, the combination of the frame-wise watermarking strategy and the Temporal Matching Module is a natural and persuasive approach to the challenges of variable-length videos and temporal attacks, demonstrating sound design rationale.
- The distinguishing feature of the proposed method is that, compared to its counterpart VideoShield, it improves video quality, robustness against temporal tampering, and invisibility. The improvement in video quality stems from the PRC-based pseudorandom Gaussian initialization, which embeds the watermark without significantly disturbing the initial noise distribution. The robustness against temporal tampering comes from embedding watermarks on a per-frame basis and using the TMM to align sequences even after frame deletion, insertion, or swapping. As for invisibility, the pseudorandom initialization makes the watermark pattern less detectable than VideoShield. The experimental results clearly confirm that each component of the proposed method contributes to performance improvements. Of course, this area appears relatively underexplored, and the improvements themselves do not seem to come from a particularly difficult domain to advance. Nevertheless, the paper demonstrates solid improvements over VideoShield, which I consider a valid contribution.

**Audience:**

Yes

**Audience Explanation:**

The paper addresses an important topic and makes a certain level of contribution to the field.

**Claims And Evidence:**

Yes

**Claims Explanation:**

While the paper contains some slightly exaggerated expressions (e.g., the claim regarding support for variable-length videos), the main claims are largely backed by experimental results, and thus I do not consider this to be a fundamental issue.

**Requested Changes:**

I would like to point out the following as weaknesses of this paper, and I believe the authors should either reflect these changes or add discussions addressing them.

- From a fair perspective, the components used in the proposed method do not appear to carry particular novelty. The work can be positioned as a practically oriented extension of existing approaches to the video domain.
- The proposed method shows strong performance under the 16-frame setting, but a clear degradation in performance is observed under the 32-frame condition. The authors attribute this to the increasing difficulty of accurately recovering the original noise via DDIM inversion as the number of frames grows, leading to larger cumulative errors. However, this performance drop should be regarded as an important weakness of the proposed method, because practical text-to-video models are likely required to generate not only short videos of around 16 frames but also videos spanning tens to hundreds of frames.
For this reason, it is necessary to first clarify whether this degradation with longer sequences is specific to the proposed method or is a common issue shared by existing in-generation video watermarking methods. In particular, (if possible) evaluating existing methods under the same 32-frame or longer condition and comparing them with the proposed method would more clearly demonstrate the scalability advantages or limitations of this approach.
Moreover, a discussion on how this issue could be mitigated in practice is also needed. For example, indicating possible solutions or future directions, such as splitting long videos into shorter segments and embedding the watermark per segment, would make the discussion of the practical applicability of the proposed method more convincing.
- There is also a concern that the temporal attack settings in the current evaluation are too simplistic. Frame dropping and frame insertion via duplication of adjacent frames appear to be attacks against which the proposed method is particularly robust. It would be necessary to also examine tasks where the robustness of the proposed method is not intuitively clear, such as substantial trimming of the beginning, end, or middle segments, playback speed changes, and more general video editing.
- The proposed method consists of multiple components, but the paper lacks an ablation study that isolates how each element contributes to robustness and accuracy improvements. Specifically, experiments are needed that individually disable the major components, namely PRC encoding, the frame-wise watermarking strategy, the random starting index, and TMM, and quantitatively evaluate their impact on performance.

---

### Review · Reviewer_Q278 · 2026-07-03

**Summary Of Contributions:**

VideoMark embeds a distortion-free watermark by sign-modulating each frame's initial Gaussian noise with PRC codewords, drawing per-frame messages from a fixed master sequence at a random start index, and extracts via DDIM inversion plus a Temporal Matching Module (TMM) that aligns the decoded sequence to the master sequence. On ModelScope (T2V) and I2VGen-XL (I2V) it reports near-perfect extraction, strong temporal/spatial robustness, and quality on par with watermark-free generation.

Strengths: clean, training-free, distortion-free initialization; frame-wise design is a sensible response to temporal attacks.
Weaknesses: validated only on obsolete 2D-VAE backbones; non-blind and non-scalable extraction; capacity/robustness claims are inflated and mutually inconsistent; several headline numbers rest on cherry-picked configs.

**Audience:**

Yes

**Audience Explanation:**

Content attribution for video diffusion is timely and relevant to TMLR readers; the failure modes here are themselves informative. The topic warrants publication once the evidence matches the claims.

**Broader Impact Concerns:**

None beyond the standard dual-use nature of watermarking/attribution; a brief statement on false-attribution risk and key management would suffice.

**Claims And Evidence:**

No

**Claims Explanation:**

1. Obsolete assumption, outdated models. The "one latent frame = one pixel frame" premise breaks on modern causal-3D-VAE models; only 2023-era 2D-VAE backbones are tested. A concurrent ICLR 2026 paper (SIGMark) empirically shows VideoMark degrades substantially on causal-3D-VAE models due to incorrect frame grouping.

2. Inflated/contradictory capacity. With a fixed master sequence, the per-video attributable payload is ~log₂(L) bits, not "512×16=8192."  High capacity and redundancy-based robustness cannot both hold, yet Figure 1 maxes both.

3. Cherry-picked configs. Extraction uses 512 bits/frame while the spatial-attack columns silently use 32 (§5.2.3). T2V scores 1.000 on every attack — and so does the baseline, indicating the attacks are too weak rather than the method flawless.

4. "Variable-length" unsupported. All main results are 16 frames; the only ablation shows I2V dropping 0.997→0.816 at 32 frames.

**Requested Changes:**

1. Results on ≥1 causal-3D-VAE model with temporal attacks intact.
2. Table 1 with a single fixed payload across all columns, and attacks strong enough to push a baseline below 1.000.
3. Report extraction latency vs. number of registered videos; clarify the true per-video attributable payload.
4. Extend length ablation well beyond 32 frames; add a "PRC-per-frame without TMM" ablation to isolate TMM's contribution.

---

### Review · Reviewer_NukN · 2026-07-10

**Summary Of Contributions:**

This paper proposes a training-free in-generation watermarking framework for video diffusion models, named VideoMark. The method embeds frame-wise messages by modulating the sign of each frame's initial Gaussian noise using PRC codewords, while aiming to preserve the marginal Gaussian distribution of the initial latent noise. To handle temporal attacks, the method samples frame messages from an extended sequence with a random starting index and applies a Temporal Matching Module (TMM) to align the decoded frame-wise messages after frame deletion, insertion, or swapping.

The motivation is clear, and the frame-wise watermarking design is a natural response to temporal attacks. However, I have several concerns about the strength of the evidence. First, the claims about variable-length robustness are not fully supported, since the main experiments are conducted on 16-frame videos and the performance already drops clearly at 32 frames (ModelScope). Second, the robustness and capacity claims are difficult to interpret because different bit lengths are used across methods and because some robustness experiments appear to use a reduced payload. Third, the temporal robustness evaluation is limited to relatively simple synthetic attacks, and the gain over the baseline in the matching accuracy comparison appears small in some cases. Finally, the method relies heavily on DDIM inversion, which may be fragile under realistic post-processing or mismatched generation and extraction pipelines. It is worth noting that the related work and novelty claims should be updated or qualified, especially regarding recent in-generation video watermarking methods such as Safe-Sora and SPDMark (Su et al., 2025; Fares et al., 2025).

**Audience:**

Yes

**Audience Explanation:**

Yes. Video watermarking for diffusion-based video generation is an important and timely topic for the TMLR audience.

**Broader Impact Concerns:**

I do not identify major additional broader impact concerns beyond the standard dual-use nature of watermarking. The work is intended to support content provenance and copyright protection, which can have positive societal value. However, the paper should briefly discuss false attribution risks.

**Claims And Evidence:**

No

**Claims Explanation:**

This paper provides evidence that VideoMark can work well under the reported 16-frame settings, especially on ModelScope and I2VGen-XL. However, the current evidence may not be sufficient to support the stronger claims about variable-length robustness, high-capacity robustness, practical robustness, and uniqueness among in-generation video watermarking methods as detailedly listed below.

- (1) The variable-length claim is somehow overstated. The main experiments are conducted on 16-frame videos. The length ablation shows clear degradation when the number of frames increases to 32 (ModelScope). This is especially important because practical T2V systems are often expected to generate videos longer than 16 frames. Therefore, the paper should not claim robust support for variable-length videos without broader length evaluations and a clearer discussion of the scalability limitation.

- (2) The capacity and robustness results are hard to interpret. Table 1 reports VideoMark with a bit length of $512\times16$ or $512\times15$, but the spatial attack evaluation is described as using only 32 bits per frame. Since the paper also shows that robustness decreases as the message length increases, it is unclear whether the near-perfect spatial robustness would still hold under the advertised full-payload setting. In addition, different methods are compared under different payload sizes. While this kind of default-setting comparison can be informative, it is not sufficient to establish robustness superiority, because payload size and robustness are usually coupled in watermarking. The paper should either provide matched-payload comparisons or report payload-robustness curves.

- (3) The temporal robustness evidence is not sufficiently convincing. The evaluated temporal attacks are limited to frame swap, insertion, and drop, which are relatively simple synthetic operations and appear favorable to the proposed frame-wise matching design. In Table 4, the matching accuracy is also close to the baseline in several cases, so the practical advantage of TMM over simpler temporal matching or resampling strategies is not fully established. More realistic temporal transformations, such as trimming the beginning, middle, or end of the video, playback speed changes, frame-rate conversion, and common video editing operations, would be needed to support the robustness claim.

- (4) The method may rely heavily on DDIM inversion for extraction. This is a strong practical assumption. In real deployment, the verifier may not know the exact scheduler, number of inference steps, VAE, guidance scale, prompt, preprocessing, or editing pipeline. Since the watermark is recovered from the inverted initial noise, small mismatches or post-processing operations may significantly affect the recovered sign pattern. One particularly relevant attack is to take a generated video, apply VAE encoding, perturb the latent representation, for example with latent-space blur, smoothing, denoising, or compression, and decode it back. Such an attack may preserve perceptual video quality while disrupting the latent sign information needed for PRC decoding. This type of latent-space post-processing or regeneration attack is not evaluated.

- (5) The empirical improvement on the T2V setting appears somewhat incremental. ModelScope is a common T2V backbone and several results are already near saturation. Some baselines also obtain very high or perfect scores under the tested attacks. This suggests that the tested attacks may be too weak to distinguish the methods, rather than proving strong robustness under realistic conditions.

- (6) The related work and uniqueness claims should be revised. The paper states that only one in-processing video watermarking method exists. This statement should be carefully qualified by the submission date and by the definition of “in-processing.” For example, Safe-Sora embeds graphical watermarks directly into the video generation process, and SPDMark proposes an in-generation video watermarking framework based on selective parameter displacement of a video diffusion model (Su et al., 2025; Fares et al., 2025).

- (7) The writing and citation formatting need improvement. In many places, citations are appended directly after method names or technical terms in a grammatically awkward way, such as “Leveraging the reversibility of DDIM Song et al. (2020)” or “Tree-Ring Wen et al. (2023).” If the citation is used only as a source reference, it should be parenthesized, such as “DDIM (Song et al., 2020)” or “Tree-Ring (Wen et al., 2023).” Alternatively, the sentence should be rewritten as a textual citation, such as “Song et al. (2020) proposed DDIM.” This issue appears throughout the paper and should be corrected.

**Requested Changes:**

- It is suggested to clarify the scope of the variable-length claim. Since the main results are on 16-frame videos and the 32-frame setting already shows clear degradation, the authors should either provide broader evaluations on longer videos or tone down the claim about variable-length robustness.

- The payload settings are suggested to be consistent and transparent. Table 1 reports $512\times16$ or $512\times15$ bits, while the spatial attack setting appears to use only 32 bits per frame. The actual payload used in each experiment should be stated, while either matched-payload comparisons or payload-robustness curves are provided.

- The current attacks are limited to frame swap, insertion, and drop. The authors should include more realistic temporal transformations, such as trimming, playback speed changes, frame-rate conversion, and common video editing operations.

- It is suggested to clarify what information is required at extraction time, such as scheduler, VAE, inference steps, guidance scale, prompt, and preprocessing. They should also evaluate robustness under mismatched extraction settings and latent-space post-processing attacks, such as VAE encode, latent blur or denoising, and VAE decode.

- Citations that are only used as source references should be parenthesized, e.g., “DDIM (Song et al., 2020)” or “Tree-Ring (Wen et al., 2023),” instead of being appended directly after method names. Similar issues throughout the paper should be corrected. Moreover, the comparisons between this paper and two works from NeurIPS 2025 and CVPR 2026 (Safe-Sora and SPDMark) are welcome.